# Promiscuous feeding on multiple adult honey bee hosts amplifies the vectorial capacity of *Varroa destructor*

Zachary S. Lamas[1,2]*, Serhat Solmaz[1,3], Eugene V. Ryabov[1,2], Joseph Mowery[4], Matthew Heermann[2†], Daniel Sonenshine[2], Jay D. Evans[2], David J. Hawthorne[1]

1 Department of Entomology, University of Maryland, College Park, Maryland, United States of America, 2 United States Department of Agriculture—Agricultural Research Service, Bee Research Laboratory, Beltsville, Maryland, United States of America, 3 Apiculture Research Institute, Ministry of Agriculture and Forestry, Ordu, Turkey, 4 United States Department of Agriculture—Agricultural Research Service, Electron & Confocal Microscopy Unit, Beltsville, Maryland, United States of America

† Deceased.

* zaclamas@gmail.com

**Data Availability Statement:** All relevant data are within the manuscript and its Supporting Information files.

## Abstract

*Varroa* destructor is a cosmopolitan pest and leading cause of colony loss of the European honey bee. Historically described as a competent vector of honey bee viruses, this arthropod vector is the cause of a global pandemic of Deformed wing virus, now endemic in honeybee populations in all *Varroa*-infested regions. Our work shows that viral spread is driven by *Varroa* actively switching from one adult bee to another as they feed. Assays using fluorescent microspheres were used to indicate the movement of fluids in both directions between host and vector when *Varroa* feed. Therefore, *Varroa* could be in either an infectious or naïve state dependent upon the disease status of their host. We tested this and confirmed that the relative risk of a *Varroa* feeding depended on their previous host's infectiousness. *Varroa* exhibit remarkable heterogeneity in their host-switching behavior, with some *Varroa* infrequently switching while others switch at least daily. As a result, relatively few of the most active *Varroa* parasitize the majority of bees. This multiple-feeding behavior has analogs in vectorial capacity models of other systems, where promiscuous feeding by individual vectors is a leading driver of vectorial capacity. We propose that the honeybee-*Varroa* relationship offers a unique opportunity to apply principles of vectorial capacity to a social organism, as virus transmission is both vectored and occurs through multiple host-to-host routes common to a crowded society.

## Author summary

*Varroa destructor* is an ectoparasitic mite implicated in historical colony losses of the managed honey bee *Apis mellifera* and is responsible for the global pandemic of Deformed wing virus (DWV). *Varroa* has long been described as a competent vector of DWV, but surprisingly little is known about its feeding and subsequent vectoring of viruses on adult honey bees. Through a series of experiments, we found *Varroa* actively switch from one

**Funding:** ZSL was funded by the PAm-Costco Fellowship. https://www.projectapism.org/ The funders had no role in study design, data collection and analysis, decision to publish, or preparation of the manuscript.

**Competing interests:** The authors have declared that no competing interests exist.

adult bee host to another to feed. Mites exhibit a large degree of heterogeneity in their host-switching behavior, with some mites switching frequently and others not nearly at all. Our results mirror an observation in other vector-host-pathogen relationships: a relatively small number of mites contributed to a majority of parasitized hosts. These high-frequency biters are most likely to not only transmit, but also acquire infectious pathogens as they switch from host to host. The ability to parasitize and infect multiple adult bees provides the best explanation to date for the maintenance and subsequent host-to-host spread of viruses among the long-lived worker bees common in these crowded and vulnerable colony populations.

## Introduction

Arthropod vectors transmit pathogens through their feeding bouts on susceptible hosts. *Varroa destructor*, an ectoparasitic mite that jumped hosts from *Apis cerana* to *Apis mellifera*, is an efficient vector of deformed wing virus, an infectious disease of honey bees that was originally only transmitted through horizontal and vertical routes from host to host [1]. Once *Varroa* was established in honey bee populations, vector-borne transmission occurred through feeding on developing bee brood and adult bees.

In 1964, Garret-Jones introduced a lasting mathematical framework to describe malaria transmission by biting mosquitoes [2, 3]. Historically used for human-mosquito-pathogen systems [4], such vectorial capacity (VC) models predict the number of infectious bites by vectors that would eventually arise from all the vectors biting a single infectious human on a day. Originally, only a few key parameters important for the continued transmission of malaria were included in the model. Since then, a legacy of host and vector-centric mathematical models have been developed to describe emerging and increasingly complex pathogen-vector-host systems [5].

The biting rate, the number of feedings made per vector per hour while switching from host to host, and the heterogeneity in this behavior, shape disease transmission epidemiology [6]. In fact, this one parameter of VC models disproportionately influences transmission [7, 8]. This is because high-frequency biting vectors are more likely to feed upon an infectious host, increasing the likelihood of acquiring and later transmitting parasites. Additionally, high-frequency biters leave behind a string of infectious hosts, forming a reservoir for other naive vectors to acquire infection. This model performs well across mosquito-borne diseases, and we wished to explore how well it explains vector-borne transmission in honey bees. We focused on the feeding rate of the vector, *Varroa destructor*, an ectoparasitic mite of the honey bee, *Apis mellifera*.

*Varroa* has a nearly cosmopolitan distribution in western honey bee populations [9] and this mite, with associated viruses, is a key suspect for large colony losses experienced globally [10]. Additional studies have linked *Varroa* and Deformed wing virus, a pathogen efficiently vectored by the mite, as drivers of honey bee losses in much of the world [11]. While much is known about *Varroa* feeding on immobile honey bee brood (larvae and pupae) [12–14], feeding patterns of *Varroa* on mobile adult bee hosts are poorly understood. Filling this knowledge gap will have broad implications in understanding disease transmission epidemiology of this economically important pest and transmission routes in a unique system that allows for horizontal transmission through both vectored and host-to-host routes. Notably, understanding the vectoring impacts of individual interactions between *Varroa* and honey bee hosts is critical for predicting dynamics and impacts at the colony and population levels.

Little is known about the feeding dynamics of *Varroa* on adult bees, partly because of difficulties tracking minute mite parasites on their hosts. *Varroa* are described as regularly leaving their original bee host after emergence from a brood cell, preferring nurse bees, and leaving adult bees prior to their death [15–17]. In observational studies, *Varroa* were observed to leave hosts in wintering clusters, suggesting they may actively switch from one host to another [18]. To date, there are no further descriptions of *Varroa* host-switching behavior on adult bees, let alone quantitative estimates of the crucial host-shifting rate, as needed for vector capacity (VC) transmission models and disease [4]. *Varroa* feeding on adult bees has been confirmed through several studies, both through the visualization of bee material inside mites and through the uptake of tagged material from experimental bees [19–21]. This established work suggests applying the biting rate as described in existing VC models may have biological foundations with *Varroa* and the honeybee.

Here we carried out a series of experiments to describe this key parameter of VC models as it applies to the honeybee-*Varroa* relationship. First, we show that *Varroa* indeed feed when they enter known feeding positions on adult honey bees and that the infectiveness of a mite depends on the viral state of previous hosts. Specifically, we used fluorescent microspheres to show that material passes in both directions between the host and vector, suggesting that *Varroa* can both transmit and acquire viruses from their adult hosts. The consequences of a *Varroa* feeding event may be dependent on the infectiousness of the previously parasitized host, and not solely because of an inherent characteristic of the individual *Varroa*. To test this, we followed *Varroa* in either infectious or naïve states. We observed their direct feeding on individual adult bees, where we found striking differences in virus levels and relative risk between treatment groups and between parasitized and non-parasitized nestmates. Finally, we measured the movement of *Varroa* among hosts to estimate the host-switching rate. In this manner, we describe the relative risk of *Varroa* feeding on virus-induced mortality, variation among mites in host-switching behavior, and transmission of virus between vectors and hosts and among hosts. We found remarkable promiscuity by feeding *Varroa*, with frequent daily switches from one bee to the next. These insights help clarify the roles played by *Varroa* in transmitting disease as well as the roles played by honey bees as reservoirs for nestmates and subsequent parasite encounters.

## Methods

### Cage design

A cage design by Evans et al. (2009) was used in all experiments for this study [22]. We used a clear plastic 16-ounce tumbler (Uline Crystal Clear Plastic Cups 16oz, S-22276) covered with a *Varroa*-proof mesh (noseum-netting) which also provided ventilation. A small insertion into the fabric lid was made with a razor blade, and a 2 ml Eppendorf tube was pushed through this insertion to serve as feeders. The tubes were perforated with a brad nail or a 5/64 drill bit, filled with water or 40% sucrose solution by weight. Trap doors were cut from the lower portion (side approximately 1x1 inch) of each cage, allowing for the removal of dead samples during trials. These holes were sealed by creating a duct tape door. Duct tape was folded back onto itself to seal all sticky portions and then cut into squares slightly larger than the hole in the cup. A strip of lab tape was used to secure the door to the cup. A handle was made by folding back a short section from one end of the lab tape onto itself, which allowed for easy closure and opening of the trap door. The cage was slid into another plastic cup to ensure no accidental escape through the trap door (Fig 1). Cages constructed in this manner allowed for the containment of both *Varroa* and bees. The cages were well ventilated, and the collection of dead samples was easy without interrupting the live samples.

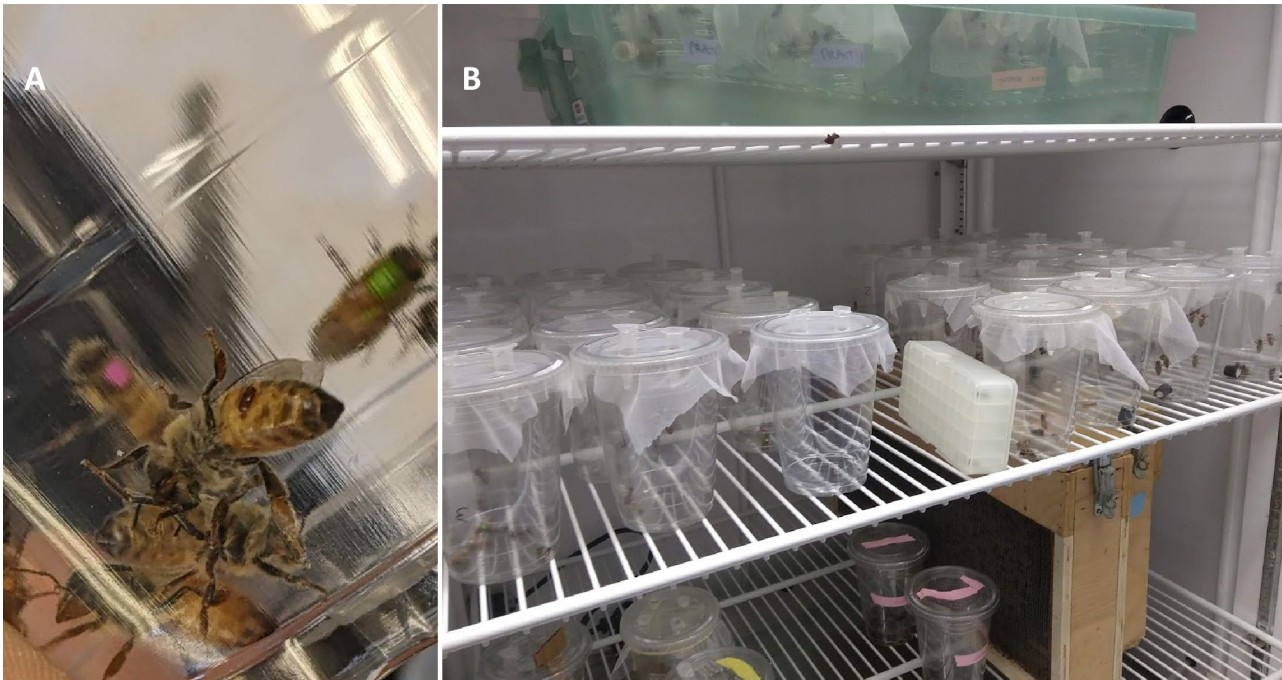

**Fig 1.** **(A)** Inside of an experimental cage. The ventral abdomen of a bee is depicted with *Varroa* visibly in feeding positions between the sternites. Bees are marked on their thorax for individual identification. **(B)** Numerous experimental cages are established and maintained inside an incubator. Noseum netting is visible and holds the sucrose feeders in place while providing ventilation and a mite-proof barrier.

## Experiments 1 and 2: Detections of feeding through microspheres

Fluorescent microspheres were used as a surrogate for bee tissue to test if Varroa were feeding on adult bees each time they entered a known feeding position. Adult nurse bees were obtained and chilled for 10 minutes at 4 degrees Celsius. Three μl of $10^7$ DAPI microspheres (Fluo-Spheres 1.0um, blue [365/415], Invitrogen, Thermo Fisher Scientific) were injected into the hemocoel between the $5^{th}$ and $6^{th}$ tergite with a 31 gauge Hamilton syringe (Hamilton Company, Reno, Nevada). Injections that showed visible dripping were rejected and not included in the study. The bees were returned to their cages and incubated for 4 hours so that the injection wounds could heal. A *Varroa* was passaged onto the bee host and left for 24 hours. After 24 hours, *Varroa* were recollected from their adult bee hosts. Their position on the bee was recorded and described as feeding or not feeding positions. Using a #5 Dumont tweezer (Montignez, Switzerland), the honey bee host was secured, and using a Chinese grafting tool (HD390, Mann Lake, Hackensack, MN), the *Varroa* was gently scooped away from its bee host. *Varroa* were anesthetized on ice for microscopy. The dorsal carapace was removed using another set of tweezers, exposing the mite's interior. The internal tissues of the *Varroa* were then smeared onto a glass slide, two μl of PBS was added, and then mounted with a cover slip. Samples were viewed under fluorescence microscopy using a Zeiss Axio Zoom V16 stereo zoom dissecting scope. Positive detections were determined by visualization of DAPI fluorescent spheres and tallied to estimate the proportion of *Varroa* that acquired microspheres from their host.

## Experiment 2: Passage of microspheres from *Varroa* to adult bee

To test if microspheres could be transferred from a *Varroa* to an adult bee via *Varroa* feeding, fluorescent microspheres were first introduced into *Varroa*. We accomplished this by having

*Varroa* feed on pupae in which three μl of $1x10^7$ DAPI microspheres in PBS buffer was injected (31 gauge needle, Micro4 microsyringe pump controller (World Precision Instruments, Sarasota, FL). A second group of pupae which served as a control was injected with PBS. The injected pupae were incubated at 34˚C for 24 hours before being fed upon by *Varroa*. Pupae showing the onset of melanization were removed from the study. *Varroa* were placed onto the injected pupae and allowed to feed for 48 hours. *Varroa* were then removed and transferred to a cage of adult bees for 24 hours. Pupae were incubated at 34˚C degrees in size-0 gel caps (Capsule Connection, Prescott, Arizona), and adult bees were incubated at 34˚C degrees in groups of approximately 40 bees in a common cage.

Bees with a *Varroa* in the feeding position were removed for dissection. Positive detections were determined by visualization of fluorescent spheres, and bees with and without microspheres were tallied to estimate the proportion of bees that acquired microspheres from a *Varroa*.

## Detection of fluorescent microspheres

To train the researcher to visualize DAPI fluorescent spheres by microscopy, a positive control of the stock solution and injected pupae were prepared on slides and then viewed by fluorescence microscopy. Z-stack images of the dorsal and ventral sides of Varroa samples were captured, and extended depth of field images were created using Zen Blue software. *Varroa* samples were then smeared on a glass slide after confirmation that microspheres were not present on the exterior of the *Varroa*.

## Experiment 3: Observation and quantification of host switching

Observations of mites switching from adult bee host to host were made across four trials in the laboratory. For all laboratory cage trials, a single frame of emerging bees was collected from healthy queen right colonies exhibiting no visible signs of disease. The frames were collected 48 hours prior to emergence and incubated at 34˚C. Newly emerged bees were collected and given a color paint mark on their thorax. Cages were made with 8 bees, individually distinguishable by their painted thorax. We utilized 7 different colors and one unmarked bee per cage. The cages were given a 40% sucrose solution and incubated for three days. At the beginning of day 4, a single *Varroa* was placed into each cage. *Varroa* were captured directly from adult bees from a single colony, then incubated on purple eye worker pupae (~ 16–17 days old) in a 0 gelatin capsules for 48 hours before transferring to the cage of workers on day 4. The presence of the *Varroa* on a host bee and which bee it was on was recorded 2 hours after introduction and every 12 hours thereafter for 15 days. In this way the number of parasitized hosts and the frequency of host switching for each *Varroa* was recorded.

It was essential in this experiment to distinguish between *Varroa* in feeding and non-feeding positions. *Varroa* in feeding positions (left, right or distal) were on the abdomen partially covered by the sternites of the bee. Non-feeding positions include the thorax or abdomen when the entire *Varroa* was visible, without any part of the *Varroa* enveloped by the bee's sternites. *Varroa* in non-feeding positions (on cage surface or in a non-feeding position on a bee) were recorded, and their movement to new hosts also recorded. Parasitized bees were those in which a *Varroa* was observed in a feeding position. Daily bee and *Varroa* mortality were recorded.

## Switching rates

A switch was considered when a *Varroa* was observed on a different bee than its previously parasitized host. The first bee a *Varroa* was observed parasitizing did not count as a switch.

Each new host subsequent to this one did. Observations were made every 12 hours during trials (+/- 2 hours). The host switching rate was calculated by dividing the total number of switches an individual mite made by the total length of time the mite existed in the trial.

## Pupae and *Varroa*

Pupae (early pink-eyed: ~ 14–15 days old) were obtained for injection by gently removing their cell capping and extracting pink-eyed pupae with a pair of soft tip tweezers. Injections were performed with a 31 gauge needle using a WPI Micro4 MicroSyringe Pump.

*Varroa* were captured along with their host bees from an infested colony. Bees with *Varroa* were placed into a cage and maintained at 34°C and 50% humidity. Pupae were removed from the comb and placed into 0 gel caps. The *Varroa* were removed individually from their honey bee host and placed in a 0 gel cap with an early purple-eyed pupal host for 24 hours. In this way, all *Varroa* collected for experiments were equalized by being on the same type of host prior to the start of the experiment.

## Experiment 4: Relative risk of *Varroa* parasitism on adult workers

We used the same cage design described previously with 8 individually marked bees to carry out this study (Evans, 2009) [22]. The bees in each cage represented a fixed population of bees which were either unchallenged or challenged by one *Varroa*. Groups challenged with a *Varroa* were further divided into groups based on the infectious status of the *Varroa*: non-infectious control, +DWV or +VDV1 (Table 1). In this way 4 groups established the study. A single *Varroa* was used in each cage replicate (n = 10 cages per group, 40 total cages). A single *Varroa* was used to reduce confounding by introducing multiple vectors within a population. The proportion of vector to host was fixed with 1 vector to 8 hosts (12.5%), a realistic infestation rate in honeybee colonies [2].

## Introduction of viral inoculum

Viral inocula (supplied by Ryabov and Evans [23,24] were injected (1 μl $10^7$ GE per μl) of inoculum in 9 μl of PBS) per pupa using Micro4 microsyringe pump controller (World Precision Instruments, Sarasota, FL). Pupae were incubated for 48 hours following injection, and then *Varroa* were introduced to these pupae by enclosing both *Varroa* and pupae in a size-0 cellulose gel cap for 72 hours. *Varroa* were then removed and placed individually into cages of 8 marked bees as described previously. *Varroa* were considered non-infectious controls if they had fed only on the PBS-injected pupae prior to the start of the experiment. *Varroa* were considered infectious if they fed upon pupae injected with viral inoculum. Because all *Varroa* in this experiment were collected from field colonies with unknown baseline levels of virus,

**Table 1. Explanation of experimental groups.** Groups, description of treatments, naming, and number of replicates in the trial. One replicate was removed from the unchallenged group, and one from the challenged + VDV1.

| Group name | Treatment | Names used in this text | Replicates in trial |
|---|---|---|---|
| Unchallenged | Bees are not exposed to *Varroa* during trial | Unchallenged, negative control group | 9 |
| Challenged | Bees are exposed to a *Varroa* that fed on a pupae injected with PBS during trial | Challenged control group | 10 |
| Challenged + DWV | Bees are exposed to a *Varroa* that fed upon a pupae injected with DWV-A inoculum prior to start of trial | *Varroa* challenged + virus group, *Varroa* challenged + DWV group | 10 |
| Challenged + VDV1 | Bees are exposed to a *Varroa* that fed upon a pupae injected with VDV1 inoculum prior to start of trial | *Varroa* challenged + virus group, *Varroa* challenged + VDV1 group | 9 |

*Varroa* in this trial harbored an unknown viral load. To account for this, we collected and treated the *Varroa* similarly for all groups. The only difference was the virus status of pupae they fed upon immediately prior to the start of the trial.

## Molecular preparation and qPCR

Total RNA was extracted from whole bees using Trizol reagent, using standard techniques [25], then RNA was used to produce cDNA using single reaction reverse transcriptase according to manufacturer specifications (BioRad, IScript). Total viral cDNA was quantified using real-time qPCR and a 10 fold dilution series of prepared standards precisely as described in Posada-Florez et al. [26].

## Statistical analysis

Data was analyzed in Rstudio using BaseR and various imported packages. In experiments 1 and 2, the frequency of microsphere presence in the parasitic *Varroa* and the host bees were tallied, and no further analysis was performed. In experiment 3, the per-day switching rate of *Varroa* (switching rate) was calculated by dividing the number of host switches by the number of days a *Varroa* persisted in the trial. When calculating the total number of parasitized hosts the first parasitized bee was included. Variation among *Varroa* in host switching rate was estimated using summary statistics. Differences in switching rates among *Varroa* was calculated by acquiring estimates of the population mean using a one-way t-test and then comparing mites by ranked groups, as well as providing descriptions of individuals which fell above and below these estimates. To assess the relationship between the number of bees parasitized by each *Varroa* over the number of days in the trial, we performed a weighted least squares by calculating fitted values from a regression and using weights of fitted values. Initial models resulted in residuals not meeting assumptions of normality. For this reason, non-parametric tests were used. These included the Mann-Whitney-U Test, Kruskal-Wallis, and weighted least squares regression.

In experiment 4, we assessed bee mortality in the four treatment groups (unchallenged, challenged control, challenged + DWV-A, challenged + VDV1) using a Kaplan-Meier survivor analysis (estimated using the survival and survminer packages in R). A log-rank test was used to compare survivorship amongst treatments. A bee or *Varroa* was considered to survive the trial when it remained alive for the whole length of the trial, which was set to 15 days.

Relative risk estimates were calculated both across treatment groups, and within groups using an unconditional maximum likelihood estimation. When making relative risk assessments across groups, risk in the unchallenged group was compared to the risk in challenged group. When making within-group relative risk assessments, risk outcomes for parasitized bees were compared with the risk assessments of their non-parasitized nestmates. In all relative risk assessments, bees were compared with counterparts that had equal exposures. Confidence intervals for these groupings were calculated using normal approximation. The Epitools package was used to calculate the relative risk estimates. Time to death (TtD) was calculated by measuring the length of time between when a bee was first observed parasitized and when first observed dead. TtD was compared across groups using an ANOVA and Tukey post hoc analysis with Bonferroni adjustments. Viral loads (DWV-A and VDV1) of bees, estimated via rtPCR, were calculated and compared across treatments, including comparing parasitized and non-parasitized bees, using a non-parametric Kruskal-Wallis ANOVA with a post hoc Dunn test.

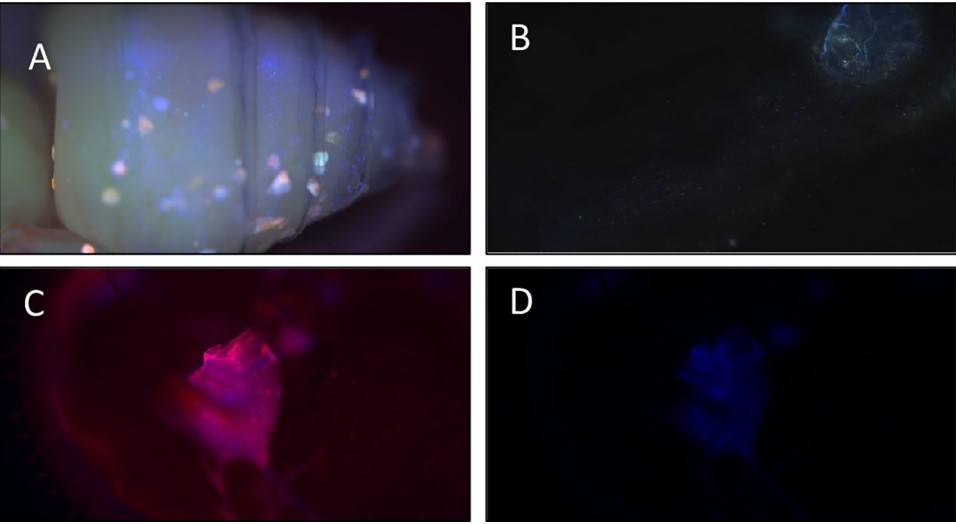

**Fig 2. A.** DAPI microspheres 1.0um, blue [365/415], Invitrogen, Thermo Fisher Scientific visualized under fluorescent microscopy through the cuticle of a worker pupae. B. DAPI FluoSpheres present in a *Varroa* which fed upon an injected adult bee. C-D The dorsal surface was partially removed to visualize the microspheres within the *Varroa*.

## Results

### Experiments 1 and 2: Bidirectional fluid movement between bee host and mite

Most *Varroa* (93.75%, 16/17 *Varroa* observed) had observable fluorescent microspheres within their digestive tract after entering feeding positions on injected adult bees. While detecting fluorescent microspheres was reliable for the movement of microspheres from bee to *Varroa*, detecting the microspheres that moved from *Varroa* to bee was less so (1/17 *Varroa* observed). Microspheres were observed freely moving within the hemocoel of the honeybee under fluorescent microscopy from the outer abdominal wall inwards, while the *Varroa* was still in a feeding position between the 3rd and 4th sternites of sampled bees. Examples of microsphere detections can be found in Fig 2.

### Experiment 3: Switching rates of *Varroa* destructor on adult worker bees

Mites showed a large heterogeneity in their host-switching behavior. *Varroa* were observed across 4 trials (N = 70). *Varroa* switched hosts every 2.5 days on average (switching rate mean ± SD = 0.369 ± 0.21 hosts/day). Time was significant, but did not account for approximately half of the variability in the number of switches made by *Varroa* over the trials (WLS regression, $R^2$ = 0.5514, $F_{1, 68}$ = 83.58, $p < 0.0001$). We accounted for longevity of the *Varroa* by dividing the number of switches a *Varroa* would make by the number of days that *Varroa* survived in the experiment (mean = 12.5 days ± SD = 3.5). The switching rate was not significantly different between trials (Kruskal-Wallis, $H^2$ = 5.697, DF [3], $p = 0.127$). The lowest frequency switching mites switched at significantly lower rates than the population mean ($t(69)$ = 10.293, $p < 0.0001$, 0.32–0.42, 95% CI). In fact, of the 70 mites within the experiment only 13 mites switched at rates within the estimated population mean. In comparison, 30 mites switched below, and 27 mites switched above estimates of the population mean (Table 2). There was no significant difference in the average switching rates of mites that survived the trial and ones which died during the trial ($p = 0.99$, Mann-Whitney U Test)

**Table 2. Mite switches and contribution to parasitized bees.** Counts of *Varroa* in the trials and the number of bees they parasitized. The percentage and cumulative percentage of parasitized bees are presented. The mean time of *Varroa* survivorship is presented here. *Varroa* switched hosts every 2.5 days on average (switching rate mean ± SD = 0.369 ± 0.21 hosts/day, 0.32–0.42, 95% CI).

| Number of *Varroa* which contributed to bites (percent of the population) | Number of individual adult bees parasitized (mean ± SD) | Percent of bees parasitized of total population | Cumulative percent of bees parasitized | Mean switching rate (SD) | Cumulative time (days) these *Varroa* were alive in the trial (mean ± SD) |
|---|---|---|---|---|---|
| 27 (38.6%) | 82 (3 ± 1) | 25% | 25% | 0.11 (0.08) | 343 (12.5 ± 3.5) |
| 17 (24.2%) | 82 (4.8 ± 1.51) | 25% | 50% | 0.37 (0.07) | 207.5 (12 ± 3) |
| 13 (18.6%) | 82 (6.3 ± 1.60) | 25% | 75% | 0.64 (0.07) | 162.5 (12.5 ± 3.5) |
| 13 (18.6%) | 84 (6.5 ± 2.40) | 25% | 100% | 0.67 (0.13) | 160.5 (12 ± 4) |

*Varroa* did not equally contribute to the number of parasitized bees in the trials. Mites that were the lowest frequency switchers contributed to fewer parasitized bees than the highest frequency switchers while, on average surviving for equal times in the study (Table 2). Time did not explain a majority of the variability in the number of bees parasitized within the trials (WLS regression, $R^2 = 0.571$, $F(1, 68) = 90.59$, $p < 0.0001$, Fig 3).

### Experiment 4: Relative risk of *Varroa* parasitism on adult worker bees

The presence of a *Varroa* among a group of worker bees was associated with increased bee mortality (Fig 4). Overall survivorship of adult bees was highest in the unchallenged group and significantly different from any of the *Varroa* challenged groups (Kaplan-Meier survival analysis, $p < 0.0001$, N = 303). Bees in the challenged groups died at faster and higher rates than bees within the unchallenged group (Kaplan-Meier survivor analysis, $p < 0.0001$, N = 303), however, there was no significant difference in survivorship of bees between any of the *Varroa*

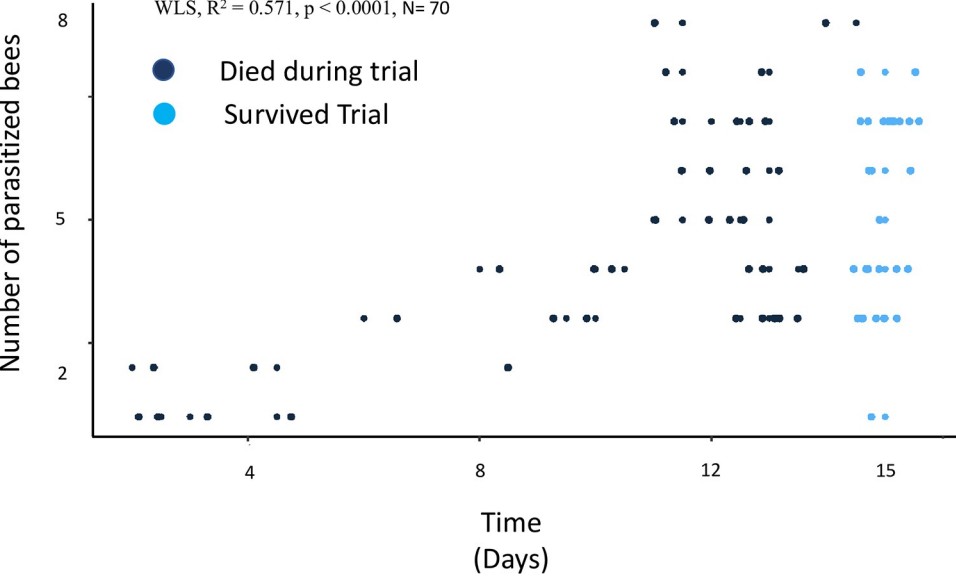

**Fig 3. Number of bees parasitized by individual *Varroa* over time Each point is a *Varroa* observed over the course of a 15-day trial as it parasitized a small group of bees in the laboratory (8 bees per group, N = 70 replicates).** The y-axis represents the number of hosts parasitized by an individual mite, while the x-axis represents how long the mite survived in the trial. Longevity was not a significant factor contributing to the number of parasitized hosts ($p = 0.124$, Mann-Whitney U Test, nor was time, which was weakly correlated. (WLS regression, $R^2 = 0.571$, $F(1, 68) = 90.59$, $p < 0.0001$) Individual points have been slightly offset on the x-axis using the jitter function in the ggplot package in R so that points are not hidden by overlapping. All points should be read to the nearest whole number on the y-axis.

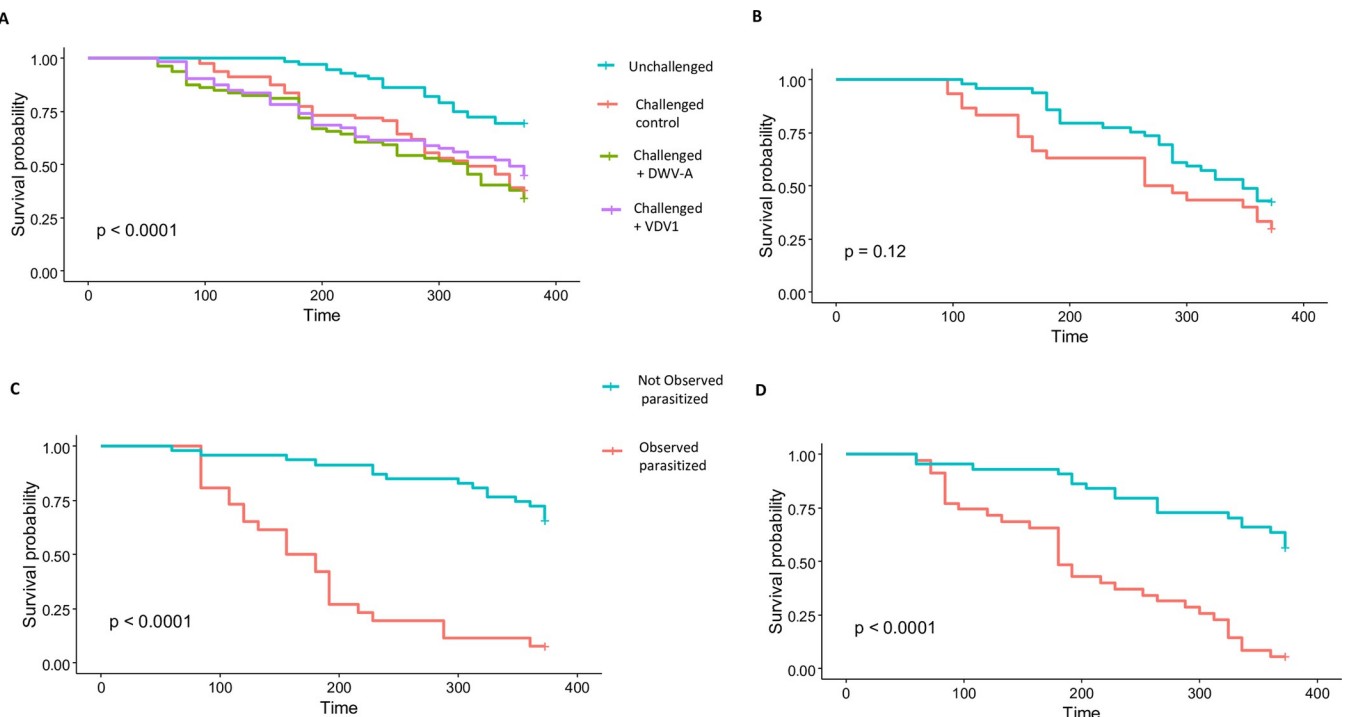

**Fig 4. Survivorship of parasitized and non-parasitized bees**: **A**. Survival analysis of bees from all treatment groups in the trial. **B, C, D** Survival analysis of parasitized and non-parasitized bees within each challenged trial group. **B.** Survivorship analysis of challenged control bees. **C.** Survivorship analysis of challenged + DWV bees. **D.** Survivorship analysis of challenged + VDV1 bees. There was a significant difference in survivorship between parasitized bees in the challenged + virus groups ($p$ <0.0001) but not in the challenged control group ($p$ = 0.12).

challenged groups whether or not an added virus was present (i.e., whether *Varroa* had fed on virus-infected pupae or on non-infected pupae prior to transferring to adult worker bees) (Pairwise Log-Rank post hoc test, $p$ = 0.33–0.6081). However, parasitism and viral treatments significantly influenced bee survivorship within the *Varroa*-challenged groups. Bees parasitized by a *Varroa* died at faster rates than their non-parasitized counterparts only within the challenged +VDV1 and challenged + DWV groups (Kaplan-Meier survival analysis, $p$ < 0.0001). There was no significant difference in survivorship between parasitized and non-parasitized individuals within the challenged control group (Kaplan-Meier survival analysis, $p$ = 0.12). The length of time from first observed *Varroa* feeding on an adult bee to death was longest in the challenged control group. Bees in this group lived for an average of 128 hours after first observed *Varroa* feeding (SD = 79 hours, n = 30). Time to death was shorter in the challenged + DWV group (96 hours) and shortest in the challenged + VDV group (87 hours). Differences were significant between the challenged control and challenged + VDV1 group, but not significantly different between the challenged control and challenged + DWV group. ($p$ = 0.027, ANOVA, Tukey post hoc, $p$ = 0.0275 challenged + VDV1 and $p$ = 0.082 challenged + DWV. *Varroa* died in the challenged + virus groups than in the challenged control group. Still, survivorship of the vector was not significantly different across any of the groups (Kaplan-Meier survivor analysis, $p$ = 0.43, N = 29).

Risk and relative risk assessments were made between treatment groups. Risk was calculated as the chance of an outcome occurring (death). Relative risk was analyzed by comparing the risk of death in one group compared to the risk of death in another treatment group. For this work, we compared the risk of death in the unchallenged group to the risk of death between each challenged group. An additional assessment of relative risk was made comparing

**Table 3. Relative risk of death between and within groups. Risk and relative risk table** Risk reported for bees within their own group (column 2). Relative risk estimates with confidence intervals reported between challenged groups to the unchallenged control (column 3). Relative risk estimates between parasitized and non-parasitized individuals within their own group (column 4).

| Group | Risk | Relative risk of death between unchallenged and challenged groups (95CI) | Relative risk of death between parasitized and non-parasitized bees (95CI) |
|---|---|---|---|
| Unchallenged | 0.29 | - | - |
| Challenged | 0.62 | 2.16 (1.47, 3.17) | 1.23 (0.87, 1.72) |
| Challenged + DWV | 0.66 | 2.29 (1.57, 3.35) | 2.18 (1.54, 3.09) |
| Challenged + VDV1 | 0.56 | 1.91 (1.27, 2.85) | 2.71 (1.79, 4.10) |

survivorship outcomes of parasitized and non-parasitized bees within their own respective group. Bees in the unchallenged group had the highest survivorship and incurred the least risk versus members of any other group (Table 3, column 2). The relative risk of death was higher and significantly different between every challenged group and the unchallenged group (2.16 challenged control, 1.91 challenged + VDV1, 2.29 challenged + DWV).

Parasitized bees in the challenged + virus groups had a high relative risk of death from a *Varroa* feeding and died shortly after being parasitized. In contrast, the non-parasitized nestmates in these groups experienced a relatively low risk of death (Table 3, column 4). However, non-parasitized bees in the challenge control group experienced a high relative risk of death. In fact, relative risk was higher for non-parasitized bees in the challenged control group than non-parasitized counterparts in either of the challenged + virus groups (challenged control: 1, Challenged + DWV: 0.76 (0.50, 1.15), Challenged + VDV: 0.60 (0.37, 0.95).

## Bee survivorship after *Varroa* feeding events

Parasitized bees continue to comingle with their non-parasitized nestmates until death or removal from a colony. In our studies the length of exposure of parasitized bees with their non-parasitized nestmates was variable between groups: highest in the challenged control group, followed by the challenged + DWV-A and challenged + VDV1 groups (Table 4). In both *Varroa* challenged + DWV-A, and challenged + VDV1 groups, observable feeding by a mite resulted in the death 94.3% and 92.9% of the time. These parasitized bees were short lived and never developed high levels (above 8 $\log_{10}$ genome equivalents (GE)) of DWV-A.

In contrast 8 parasitized individuals in the challenged + VDV1 group had high levels of VDV1, all of which died during the 15-day trial. However, non-parasitized individuals still represented the majority of high VDV1 infections within this group, with 14 of the 22 most infectious individuals in the challenged + VDV1 group being non-parasitized, of which only 2 died prior to the end of the trial. (Tables 4 and 5 for descriptive statistics and count data)

**Table 4. Mean time to death after a *Varroa* feeding.** Mean time (hours) to death reported for bees within each challenged group for the whole trial, followed by count data of the number of bees parasitized in the trial. Analysis is provided in the final column of the percentage of parasitized to non-parasitized bees per cage within each group (±SD).

| Group | Number of Parasitized bees (total bees in trial) | Mean Time to death (hours) after *Varroa* feeding(SD) |
|---|---|---|
| *Varroa* Challenged | 30 (72) | 128 (79) |
| Challenged + DWV | 35 (79) | 96 (46) |
| Challenged + VDV1 | 26 (73) | 87 (24) |

**Table 5. Counts of parasitized and non-parasitized bees during experiment 4.** Count data provided for parasitized and non-parasitized bees in Experiment 4.

| Group | Non-Parasitized bees that survived | Non-Parasitized bees that died | Parasitized bees that survived | Parasitized bees that died |
|---|---|---|---|---|
| Unchallenged | - | - | - | - |
| Challenged | 21 | 28 | 9 | 21 |
| Challenged + DWV | 25 | 19 | 2 | 33 |
| Challenged + VDV1 | 31 | 16 | 2 | 24 |
| Group | Non-parasitized bees (survived) with high levels of DWV-A (VDV1) | Non-parasitized bees (died) with high levels DWV-A (VDV1) | Parasitized bees (survived) with high levels DWV-A (VDV1) | Parasitized bees (died) with high levels DWV-A (VDV1) |
| Unchallenged | - | - | - | - |
| Challenged | 0 (0) | 1 (0) | 2 (0) | 1 (0) |
| Challenged + DWV | 1 (0) | 4 (0) | 0 (0) | 0 (0) |
| Challenged + VDV1 | 1 (12) | 1 (2) | 0 (0) | 0 (8) |

## Viral loads across groups and between bees within groups

Viral loads differed significantly across groups and between parasitized and non-parasitized bees within their respective groups. There was a significant difference in viral loads across groups ($p < 0.0001$, Tables A and B in S1 Tables.) DWV-A levels were lowest in the unchallenged group and significantly different between bees within all challenged groups (post-hoc Dunn test, $p < 0.0001$). DWV-A levels were highest in the challenged control group and were significantly higher than in the two other challenged groups + virus (post-hoc Dunn's test, $p < 0.0001$). DWV-A loads were not significantly different between the two challenged + virus groups (post-hoc Dunn's test, $p = 0.61$). Surprisingly, despite dying quickly after a mite feeding, parasitized bees in the challenged + virus groups failed to develop high levels of DWV-A infection (Table 6). VDV1 levels were highest in the challenged group + VDV1 and significantly higher than any other group ($p < 0.0001$, Kruskal-Wallace, df = 4, post-hoc Dunn's test, $p < 0.0001$). There was no significant difference between VDV1 levels and any other group in the trial.

## Viral loads in non-parasitized bees

There was a significant difference in DWV-A viral loads per bee across groups (Kruskal-Wallis $H^2 = 35.255$, df = 3, $p < 0.0001$). Non-parasitized bees developed high levels of DWV-A

**Table 6. Mean viral loads DWV-A and VDV1 across experimental groups (log$^{10}$ GE per bee).** Means ± SD for DWV-A and VDV1 viral loads provided in order from the group and within parasitized and non-parasitized cohorts. Analysis can be found in subsequent Tables A and B in S1 Tables.

| Group | Mean DWV-A loads (SD) | Mean VDV1 loads (SD) | Mean DWV-A loads non-parasitized bees (SD) | Mean DWV-A loads parasitized bees (SD) | Mean VDV1 loads non-parasitized bees (SD) | Mean VDV1 loads parasitized bees (SD) |
|---|---|---|---|---|---|---|
| *PreTrial Collection* | 3.74 (0.62) | 6.45 (0.14) | - | - | - | - |
| *Unchallenged* | 5.20 (1.68) | 5.40 (1.42) | - | - | - | - |
| *Challenged* | 4.47 (1.72) | 5.41 (1.48) | 4.91 (1.44) | 5.70 (1.94) | 5.44 (1.48) | 5.32 (1.34) |
| *Challenged + DWV* | 3.68 (0.44) | 5.06 (1.45) | 4.63 (2.09) | 4.25 (0.97) | 5.36 (1.53) | 5.49 (1.43) |
| *Challenged + VDV1* | 4.13 (1.22) | 7.17 (1.64) | 4.23 (1.46) | 3.93 (0.52) | 7.35 (1.53) | 6.83 (1.81) |

infection ($> 8 \log_{10}$ GE per bee) in all of the *Varroa-challenged* groups. Non-parasitized bees in the challenged control group and the challenged + DWV group had significantly higher levels of DWV-A than bees in the unchallenged group (Dunn post hoc test, $p < 0.0001$ and $p = 0.046$). There was no significant difference in DWV-A levels between non-parasitized bees in the challenged + VDV group and bees in the unchallenged group (Dunn post hoc test, $p = 0.053$). VDV1 levels were significantly different when compared across all groups (Kruskal-Wallis $H^2 = 71.774$, df = 4, $p < 0.0001$). VDV1 levels were highest in non-parasitized bees in the challenged + VDV1 group, and significantly different compared to non-parasitized bees in all other groups (Dunn post hoc test, $p < 0.0001$). There were no significant differences between VDV1 levels of any other group (Table 6).

## Viral loads in parasitized bees

Parasitized bees, bees in which a *Varroa* was observed in feeding position at least once during the trial, only developed high levels of DWV-A infection in the challenged control group and not within the challenged + virus groups (Table 6). Parasitized bees with high levels of DWV-A represented a minority of all bees that developed high infection levels: 3 out of 11 bees. DWV-A levels were significantly higher for parasitized bees in the challenged control group than in any other group (Fig 5). VDV1 levels were highest in the challenged + VDV1 group (Table 6). There was no significant difference in viral levels between parasitized or non-parasitized bees within the challenged + VDV1 group. However, there were more observations of non-parasitized bees developing high DWV-A infection levels than their parasitized counterparts. Within the Challenged + VDV1 group, 22 bees developed high levels of VDV1. Of those bees, only 8 were parasitized, which all died during the 15-day trial. The majority,14, were non-parasitized, of which only 2 died prior to the end of the trial.

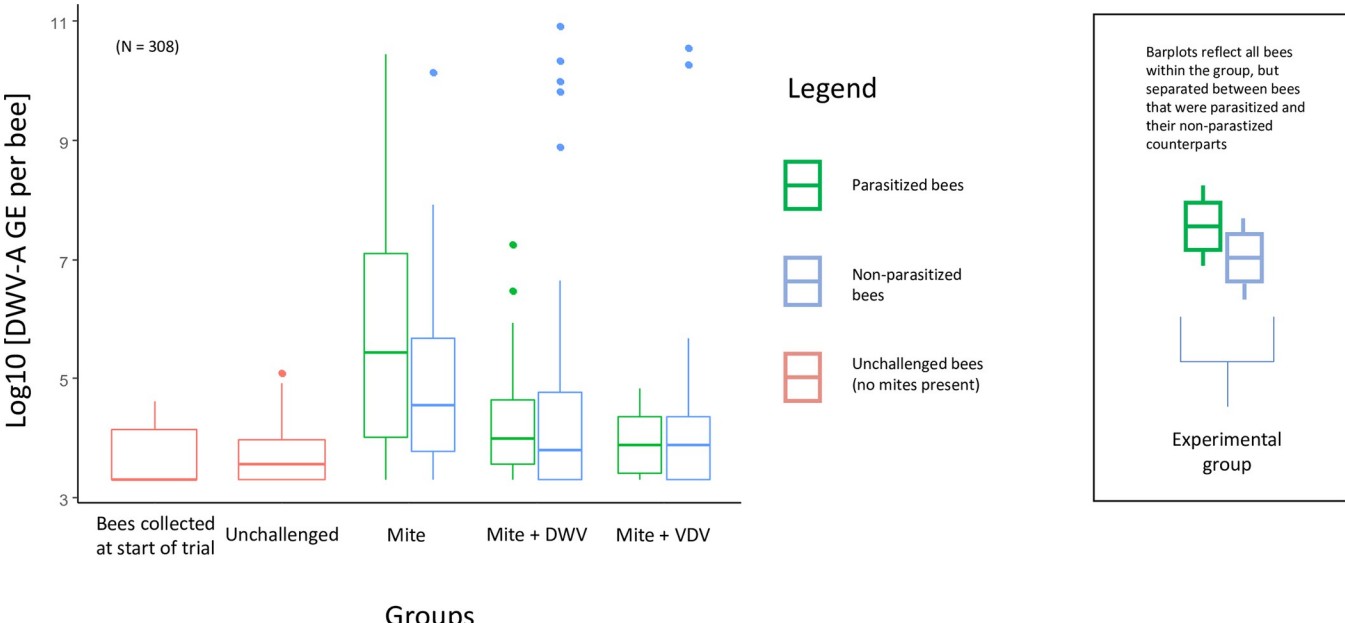

**Fig 5. DWV-A genome equivalents (GE) per bee loads of 308 individual bees sampled from experiment 4.** The four treatment groups are shown, as well as an additional group (furthest left) of bees collected at the start of the trial. For the 3 *Varroa* challenged groups, dual boxplots were used to display DWV-A GE per bee separately for parasitized and non-parasitized bees. The relative proportion of parasitized to non-parasitized bees can be found in **Table 5** and AOV analysis from **Tables A-D in S1 Tables**.

## Discussion

The biting rate is an influential parameter in VC models of mosquitoes [2]. Heterogeneity in this behavior has overwhelming responsibility for driving pathogen transmission in mosquito-borne diseases [6]. Through a series of experiments, we constructed the biological and behavioral framework suggesting that the biting rate, as it is used in VC models with mosquitoes, could be applied to the honeybee-*Varroa* relationship. Our experiments provide quantitative estimates for the host-switching rates of *Varroa* mites from one adult bee to another and the impacts of those switches on the survivorship of their adult bee hosts and disease vectoring. Varroa's consistent acquisition of fluorescent microspheres when feeding on adult bees shows that host-switching is best seen as a pursuit of feeding on adult bees, not simply seeking resting places or evading hygienic grooming. *Varroa* primarily feed on the fat body of adult honey bees, while incidentally ingesting hemolymph [19, 27]. Regular uptake of the microspheres in our trials also confirmed ingestion of free-floating material, suggesting virus particles distributed throughout the hemocoel could be acquired independently of their presence in fat body. We observed fluorescent microspheres moving bidirectionally between vector and host, suggesting that *Varroa* can acquire infectious material from one host and pass that material onto a subsequent host. Bidirectional movement of tissues and fluid between vector and host implies that *Varroa* can acquire infectious material from one host and potentially pass that material onto the next host. *Varroa* are described as a mechanical vector for DWV-A and lose their ability to transmit the virus when passaged upon a series of non-infectious hosts [12], though this might not be a case for VDV1 (DWV-B) [28]. Our work coupled with previous work, suggests *Varroa* infectiousness may partly depend on the host condition they most recently parasitized [29, 30]. The frequency at which a mite switches from one adult host to another to feed could shape transmission in a honeybee population. In mosquitoes, the highest frequency biters are the individuals most likely to transmit a pathogen. They are also the most likely to bite an infectious host, thus acquiring the pathogen. The bidirectional material movement in our study suggests that a similar phenomenon may occur in *Varroa*. Like mosquitoes, we observed low and high-frequency biters in the *Varroa* population exhibiting significant heterogeneity in this behavior.

*Varroa* are promiscuous feeders on adult bees and express significant heterogeneity in the host-switching rate. *Varroa* which engaged in the highest frequency switches, were responsible for nearly three times as many parasitized hosts as their lower switching counterparts. For example, some *Varroa* switched 12–15 times in 15 days, returning to previously fed upon hosts because all non-parasitized bees had been exhausted. Meanwhile, slower switching counterparts switched only once in the same 15-day period, meaning that most bees in that cohort were not bitten. Like mosquitoes, high-frequency switchers would be more likely to feed upon an already infectious adult bee than a slow switching counterpart. After becoming infectious, these *Varroa* would be responsible for the most significant proportion of infected hosts, thus increasing the risk to all other *Varroa* that feed upon an infectious host and the hosts themselves. The underlying mechanism driving heterogeneity in this behavior was not studied but warrants future research. Behavioral heterogeneity could be explained by genetic differences in the *Varroa* population, whether *Varroa* had already produced or are callow daughters, or how long *Varroa* have been in the dispersal stage. Our studies attempted to limit the heterogeneity in the host population so that we could observe differences in *Varroa* behavior without confounders. In an actual bee population of mixed ages, phenotypes, and sexes, there would likely be an interaction between the behaviors of the vector and the availability or unavailability of ideal hosts. The host-switching rate may also be affected by similar factors that influence the amount of time *Varroa* spend on adult bees, such as host condition and brood availability,

which have already been shown to affect the amount of time *Varroa* spend in their dispersal phase [31].

There are apparent costs and benefits for estimating the host-switching behavior and consequences of their feeding on adult bees in laboratory settings. Here, we used a fixed host population size since both basic reproductive rate($R_0$) and vector capacity models utilize fixed populations in their estimates [2, 32]. Artificial arenas reduced the number of confounders normally inherent in a honeybee colony as cage designs eliminate many key characteristics of a honeybee colony [22] while it also reduces the degree of field relevance [22]. However, this allowed us to study the relative risk of direct feeding on adult bees and the conferred harm to nestmates without confounders and survivorship bias inherent in colony settings. Because of the social organization of a honeybee colony, it's possible that the conferred harm we observed in our trials to non-parasitized bees would not be observable in colony states where there are an ample number of newly emerged bees. A field study that tried to answer this question would be affronted by numerous confounding variables such as many *Varroa* with varying degrees of infectiousness, unobserved parasitism, cannibalized pupae as a vector for honeybee viruses, and survivorship bias from death and removal by nest mates.

Observed feeding by *Varroa* was a significant predictor of bee mortality in our trials, but *Varroa* feeding could only partially explain bee deaths. The mean time to death was significantly shorter after a *Varroa* bite in the challenged + virus groups compared to a *Varroa* bite in the challenged control group. Contact rates between non-parasitized and parasitized nestmates were longest in the challenged control group. Our results suggest that long-lived parasitized bees confer a risk of death and viral transmission to non-parasitized nestmates. If true, trophallaxis or the oral exchange of food between nestmates may serve as a more impactful route for viral transmission than currently appreciated. Relative risk was higher for non-parasitized bees in the challenged control group than for non-parasitized counterparts in the viral challenged groups. This is likely possible because bees in this group lived for extended periods after a *Varroa* feeding, giving more opportunities for contact and trophallactic interactions with non-parasitized nestmates. Our data suggest parasitized bees that died quickly after *Varroa* feeding may confer protection to non-parasitized nestmates by limiting opportunities for host-to-host transmission. In contrast, long-lived survivors may elevate risk to nestmates.

Continued research is warranted to understand how oral and contact transmission affects virus transmission dynamics in a honeybee colony. We suggest that these asymptomatic, non-parasitized bees may be responsible for the maintenance of the pathogen and potentially serve as a reservoir of infectious bees and continued viral transmission in a dense honeybee colony. We draw this hypothesis from the results of this experimental study and upon similar phenomena observed in other disease systems, namely emerging viruses that are both horizontally spread between hosts and vectored borne. Like social bees, birds that received West Nile virus or Tembusu virus through non-vectored communicable routes developed high levels of infection and lived longer than parasitized or experimentally injected subjects [33–35]. A recent study confirmed this alarming trend. Older, asymptomatic ducks shed high levels of virus to flock mates, supporting the role "supershedders" may have in an epidemic. [36] In the honeybee colony, the production of supershedders may be produced by the continual production of parasitized bees and susceptible individuals that trophallaxis with them. It is quite possible that this circulation between vector-host and host-host transmission could increase the risk of naïve *Varroa* acquiring infectious levels of DWV as they jump from bee to bee.

Continued research is needed to understand the impacts of this economically important pest on adult bees. *Varroa* switching from one adult bee to another to feed would jump trophallaxis networks which are carefully structured to maintain cohesion in the colony [37]. Not only could individual bees be connected due to a lineage of *Varroa* feedings, but entire social

networks within the colony could be bridged [38]. These social networks, which naturally exhibit degrees of independence from each other [37], would be connected via promiscuous vectors. Prolific switching by vectors would also mean the infestation rate, often measured as a proportion of *Varroa* in a sample of bees [39], would not reflect the gross number of bees fed upon. In short, more bees could have been fed upon at any given time than the total number of *Varroa* in the colony. Finally, *Varroa*, DWV, and the honeybee offer a unique relationship in which to apply vectorial capacity principles as the relationship offers multiple communicable modes of transmission, not just vectored routes. Vectoring of DWV by *Varroa* is also an evolutionarily recent phenomenon, where mathematical analysis would help describe co-adaptation by vector, pathogen, and host over time.

## Supporting information

**S1 Tables. Tables comparing viral analysis within and between groups.**
(DOCX)

**S1 Data. Data from experiment 3.**
(CSV)

**S2 Data. Survival data of hosts from experiment 4.**
(CSV)

**S3 Data. Survival data from vectors experiment 4.**
(CSV)

**S4 Data. Viral analysis data from experiment 4.**
(CSV)

## Author Contributions

**Conceptualization:** Zachary S. Lamas, Matthew Heermann.

**Formal analysis:** Zachary S. Lamas, Jay D. Evans.

**Funding acquisition:** Zachary S. Lamas.

**Investigation:** Zachary S. Lamas, Serhat Solmaz, Joseph Mowery.

**Methodology:** Zachary S. Lamas, Daniel Sonenshine.

**Resources:** Daniel Sonenshine.

**Supervision:** Jay D. Evans, David J. Hawthorne.

**Writing – original draft:** Zachary S. Lamas, Jay D. Evans, David J. Hawthorne.

**Writing – review & editing:** Zachary S. Lamas, Eugene V. Ryabov, Joseph Mowery, Daniel Sonenshine, Jay D. Evans, David J. Hawthorne.

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
