## [Decision Letter · Decision Letter 0]

27 Aug 2022

Dear Mr Lamas,

Thank you very much for submitting your manuscript "Promiscuous feeding on multiple adult honey bee hosts amplifies the vectorial capacity of Varroa destructor" for consideration at PLOS Pathogens. As with all papers reviewed by the journal, your manuscript was reviewed by members of the editorial board and by several independent reviewers. The reviewers appreciated the attention to an important topic. Based on the reviews, we are likely to accept this manuscript for publication, providing that you modify the manuscript according to the review recommendations.

This manuscript does two things very well: first, it uses the concept of vectorial capacity to describe transmission of a pathogen among honeybees by Varroa destructor; and second, in a series of experiments, it measures vectorial capacity. I found the paper to be engaging, interesting and new.

As someone who does not know a lot about this particular pathogen system, I thought an introductory paragraph about the system would perhaps be more effective than one that launches directly into a discussion of vectorial capacity.  I have made a few suggestions about that the discussion. 

Sincerely,

David L. Smith, Ph.D.

Guest Editor

PLOS Pathogens

Kirk Deitsch

Section Editor

PLOS Pathogens

Kasturi Haldar

Editor-in-Chief

PLOS Pathogens

orcid.org/0000-0001-5065-158X

Michael Malim

Editor-in-Chief

PLOS Pathogens

orcid.org/0000-0002-7699-2064

With respect to vectorial capacity:

1. The formula for vectorial capacity in Garrett-Jones 1964 paper was extracted directly from Macdonald's 1952 paper, which introduced Ro into malaria epidemiology (and, indeed, into mathematical epidemiology). Since a lot of the work was done by Macdonald, I suggest also citing Macdonald (Ref: Macdonald, G. (1952). The analysis of equilibrium in malaria. Trop. Dis. Bull. 49, 813–829.)

2. I think it's more accurate to say that it has been applied to "human-mosquito-pathogen" systems. Vectorial capacity predicts the number of infectious bites by vectors (not vectors) that would eventually arise from all the vectors blood feeding on a single infectious human on a day.

3. In the opening paragraph, there is a reference to a figure but no number was given. 

4. The set of refs 3-5 at the end of the first paragraph is a bit eclectic. I suggest (with some shame) citing a systematic review. Full disclosure, I was an author on this systematic review. (The reference is: Reiner RC, et al. (2013). A systematic review of mathematical models of mosquito-borne pathogen transmission: 1970-2010. J. R. Soc. Interface 10, 20120921.)

5. The discussion that follows about sensitivity to parameters seemed a bit confusing to me. In discussing "this one parameter of VC models" that has a disproportionate influence on transmission, the authors cite a paper that emphasizes mosquito mortality / lifespan. I think the authors are more interested in the feeding rate. The history of this idea is a bit involved, so I will elaborate: Macdonald's other 1952 paper made the cases that mosquito longevity was critically important for transmission (Macdonald, G. (1952). The analysis of the sporozoite rate. Trop. Dis. Bull. 49, 569–586.). The paper launched new research, and Macdonald revisited this idea in a paper published in 1955 about eradication. Measuring longevity has become a sort of holy grail in studies of pathogen transmission by mosquitoes. In this vector-transmitted pathogen system that the authors are describing, where the main mode of transmission is mechanical (if I understand it correctly), the factor that Macdonald was discussing (surviving the parasite's EIP) is not relevant. In this system, the authors are emphasizing that the feeding rate (in particular, host switching) does, indeed, affect transmission in at least two ways. (In mosquitoes, since blood is provisioned into eggs, which are laid and hatch into adults, the blood feeding rate could be assigned a third effect, at least for purposes of weighing vector control. I'm not sure if it applies to this system.). The authors would not be the first to draw attention to the blood feeding rate, but IMO, the paper by Novoseltsev et al. is chasing the argument about lifespan. It was Garrett-Jones, not Macdonald, who drew a bit more attention to the human blood feeding rate, especially when looked at in the context of his other paper from 1964 on the human blood index. I would suggest the authors emphasize that this paper focuses on the feeding rate and host switching (in this case), and not one that focuses on vector mortality. Not many papers on pathogen transmission by vectors have put such an emphasis on the feeding rate, which increases the novelty.

Reviewer Comments (if any, and for reference):

Reviewer's Responses to Questions

**Part I - Summary**

Reviewer #1: Overall, I enjoyed this paper and would hope to see it published. I think the mix of vector-transmission and between-host transmission of important bee viruses is a fascinating topic. The result of parasitized bees which died quickly after Varroa feeding may be conferring protection to non-parasitized nestmates by limiting opportunities for host to host transmission, whereas long lived survivors may elevate risk to nestmates, is particularly interesting. With my modeller hat on, and associated biases, I would encourage these authors to engage with vector-borne disease modellers. There is already scope for some basic first-principles analyses using a Ross-Macdonald framework; and, the interaction might even foster ideas for new experiments.

Reviewer #2: The study provides new insight on Varroa host-switching behavior on adult bees. The study is of importance and interest. The study provides interesting results on virus level in bees that relate to parasitism by Varroa on the host or not. I think that the paper can benefit from improved introduction and the same references can be used in discussion.

**Part II – Major Issues: Key Experiments Required for Acceptance**

Reviewer #1: (No Response)

Reviewer #2: The experiments were well performed.

**Part III – Minor Issues: Editorial and Data Presentation Modifications**

Reviewer #1: Mostly, the readability of the results needs improving. 5 multi-part figures and 12 tables is inundating. The authors have done a huge amount of work and I applaud them for packing it together into a single paper rather than 3 smaller papers, but some effort is needed to make it more manageable. Please see below some more specific comments.

Figure 1.3 shows the number of bees parasitized by individual Varroa over time. It distinguishes Varroa which died from those that survived the 15d trial by colour. The authors accounted for longevity of the Varroa by dividing the number of switches a Varroa would make by the number of days that Varroa survived in the experiment. I think it is this that is displayed in Fig1.3 (because some data points have non-integer values on the y-axis and a mite cannot parasitize 0.3 of a bee). But I don’t know how some Varroa on day 15 had parasitized 8 bees (i.e., 120 bees parasitized divided by the 15 days survived). I think I've misinterpreted this figure and associated text. Some more explicit description of exactly what is plotted in Fig1.3 would be appreciated.

Table 1.3 and associated text: after reading through this 3 times I gave up. I sympathise that it is a complicated set of comparisons being made between parasitised and non-parasitised bees within the same treatments versus non-parasitised bees between different treatments etc., but, I think the ms would benefit from the authors coming up with a better way of encapsulating exactly which comparisons yielded which result. Would there be a pictorial way of depicting the comparisons and combining the results from multiple tables (lessening the tirade of tables)?

Table 1.4: the final column header is ‘Mean percentage of non-parasitized to parasitized nestmates’ but the table legend states details are provided for ‘the average ratio of parasitized to non-parasitized bees in each group’. One of these two depictions is wrong. Why not present raw numbers with prevalence of parasitism in brackets afterwards?

Reviewer #2: Abstract – what does mean „now endemic in honeybee populations” – it appears misleading

Importantly, DWV can be identified in other bee species – thus, you should provide exact information

Introduction

It can be recommended: the manuscript could benefit from including information about pathways that Varroa affects. To some extent, there are similarities to ticks. Ticks are mites and thus are more related to Varroa than mosquitoes. The pathways can also be similar.

It is important to introduce, that the mite affects bees. In addition, the mite can transmit also other viruses, not only deformed wing virus. For instance, acute paralysis virus it is of high importance.

Further, the mite strongly affected virome of honey bees – immortally, the virome of Varroa naïve bees (Varroa was until now absent in Australia) is different than in other countries. Thus, information related to this can be recommended to introduce. At page 4 “with associated viruses” – it is superficial information without any reference. Authors should rather provide information which viruses Varroa transmits, which viruses were detected in the mite to be potentially transmitted to bees. In addition, virome of honey bees is of importance, because the mite then transmits viruses related to the particular colonies / sites.

Page 4 “North America” – why only North America? – the mite shifted to European honey bees in Europe much earlier and then it spread worldwide – most recently to Australia.

Page 4 “a pathogen efficiently vectored by the mite” it is question if the virus is “only” transmitted by the mite or if it replicates in the mite. Study that supports “only” transmission can be provided.

Page 5 “Varroa feeding and development on immobile honey bee brood (larvae and pupae),” – the development of bees and pathways is affected and some reference is missing.

Further, you should consider provide studies that detected DWV in other species that do not host Varroa.

Page 5 – “let alone quantitative estimates of the crucial host-shifting rate, as needed for vector capacity (VC) models of transmission and disease” there were analyzed adult mites – references can be provided. The viruses were identified using PCR, but some viruses were also identified in Varroa at protein level using proteomics.

Page 6 – “on the viral state of previous hosts.” – virome studies of honey bees can be used

You used incola of DWV-A and VDV1 – which viruses (DWV strains) predominate in Varroa parasitized bees? – adult bees? Virome studies can be used.

It appears that bees die earlier if bees are challenged to Varroa and DWV or VDV1 – but Varroa transmits also viruses – which viruses were in the mite?

Could you provide more information/discussion about presence of viral strains in bees? Some studies are available. It is of importance that controlling of Varroa infestation can affect occurrence of DWV strains only to some extent.

risk of naïve Varroa - ? not clear, could you please explain – does it mean Varroa without virus? Or bees naïve to Varroa

Minor corrections:

correct “population.-The proportion”

PLOS authors have the option to publish the peer review history of their article (what does this mean?). If published, this will include your full peer review and any attached files.

Reviewer #1: **Yes: **Laith Yakob

Reviewer #2: No

Figure Files:

Data Requirements:

Reproducibility:

References:

---

## [Editor Report · Decision Letter 1]

17 Nov 2022

Dear Mr Lamas,

Thank you very much for submitting your manuscript "Promiscuous feeding on multiple adult honey bee hosts amplifies the vectorial capacity of Varroa destructor" for consideration at PLOS Pathogens. As with all papers reviewed by the journal, your manuscript was reviewed by members of the editorial board and by several independent reviewers. The reviewers appreciated the attention to an important topic. Based on the reviews, we are likely to accept this manuscript for publication, providing that you modify the manuscript according to the review recommendations.

The authors are encouraged to revise their manuscript one more time, paying close attention to the descriptions in the text of every quantity that is discussed. In particular, the first time any quantitative term is introduced in text, the authors should take great care to describe each term in detail, so that when the shorter phrase is used later, there is no doubt about what the quantity describes. In particular, we invite the authors to modify how they describe the term`relative risk.`

Sincerely,

David L. Smith, Ph.D.

Guest Editor

PLOS Pathogens

Kirk Deitsch

Section Editor

PLOS Pathogens

Kasturi Haldar

Editor-in-Chief

PLOS Pathogens

orcid.org/0000-0001-5065-158X

Michael Malim

Editor-in-Chief

PLOS Pathogens

orcid.org/0000-0002-7699-2064

Reviewer Comments (if any, and for reference):

Figure Files:

Data Requirements:

Reproducibility:

References:

---

## [Editor Report · Decision Letter 2]

12 Dec 2022

Dear Mr Lamas,

We are pleased to inform you that your manuscript 'Promiscuous feeding on multiple adult honey bee hosts amplifies the vectorial capacity of Varroa destructor' has been provisionally accepted for publication in PLOS Pathogens.

Best regards,

Kirk W. Deitsch

Section Editor

PLOS Pathogens

Kirk Deitsch

Section Editor

PLOS Pathogens

Kasturi Haldar

Editor-in-Chief

PLOS Pathogens

orcid.org/0000-0001-5065-158X

Michael Malim

Editor-in-Chief

PLOS Pathogens

orcid.org/0000-0002-7699-2064
---

## [Editor Report · Acceptance letter]

19 Dec 2022

Dear Dr. Lamas,

We are delighted to inform you that your manuscript, "Promiscuous feeding on multiple adult honey bee hosts amplifies the vectorial capacity of Varroa destructor," has been formally accepted for publication in PLOS Pathogens.

Best regards,

Kasturi Haldar

Editor-in-Chief

PLOS Pathogens

orcid.org/0000-0001-5065-158X

Michael Malim

Editor-in-Chief

PLOS Pathogens

orcid.org/0000-0002-7699-2064